# Overparameterisation and worst-case generalisation: friend or foe?

**Aditya Krishna Menon, Ankit Singh Rawat & Sanjiv Kumar**
Google Research
New York, NY
`{adityakmenon,ankitsrawat,sanjivk}@google.com`

## Abstract

Overparameterised neural networks have demonstrated the remarkable ability to perfectly fit training samples, while still generalising to unseen test samples. However, several recent works have revealed that such models' good *average* performance does not always translate to good *worst-case* performance: in particular, they may perform poorly on subgroups that are under-represented in the training set. In this paper, we show that in certain settings, overparameterised models' performance on under-represented subgroups may be improved via post-hoc processing. Specifically, such models' bias can be restricted to their classification layers, and manifest as structured prediction shifts for rare subgroups. We detail two post-hoc correction techniques to mitigate this bias, which operate purely on the outputs of standard model training. We empirically verify that with such post-hoc correction, overparameterisation can improve average and worst-case performance.

## 1 Introduction

Overparameterised neural networks have demonstrated the remarkable ability to perfectly fit training samples, while still generalising to unseen test samples (Zhang et al., 2017; Neyshabur et al., 2019; Nakkiran et al., 2020). However, several recent works have revealed that overparameterised models' good *average* performance does not translate to good *worst-case* performance (Buolamwini & Gebru, 2018; Hashimoto et al., 2018; Sagawa et al., 2020a;b). In particular, the test performance of such models may be poor on certain subgroups that are *under-represented* in the training data. Worse still, such degradation can be exacerbated as model complexity increases. This indicates the unsuitability of such models in ensuring *fairness* across subgroups, a topical concern given the growing societal uses of machine learning (Dwork et al., 2012; Hardt et al., 2016; Buolamwini & Gebru, 2018).

Why does overparameterisation induce such unfavourable bias, and how can one correct for it? Sagawa et al. (2020a) demonstrated how such models may fit to *spurious correlations* that explain under-represented samples, which can generalise poorly. Sagawa et al. (2020b) further posited that overparameterised models have an inductive bias towards *memorising* labels for as few samples as possible, which are invariably those from under-represented subgroups. To mitigate such bias, existing approaches include subsampling majority subgroups (Sagawa et al., 2020b), and modifying the training objective (Sagawa et al., 2020a; Nam et al., 2020; Zhang et al., 2020; Goel et al., 2020). This suggests two important points regarding overparameterised models' performance:

(a) with standard training, increasing model complexity exacerbates degradation on rare subgroups;

(b) controlling this degradation may require alternate training objectives or procedures.

In this paper, we establish that while overparameterised models are biased against under-represented examples, in certain settings, such bias may be easily corrected via *post-hoc processing* of the model outputs. Specifically, such models' bias can be largely restricted to their classification layers, and manifest as structured shifts in predictions for rare subgroups. We thus show how two simple techniques applied to the model outputs — classifier retraining based on the learned representations, and correction of the classification threshold — can help overparameterised models *improve* worst-subgroup performance over underparameterised counterparts. Consequently, even with standard training, overparameterised models can learn sufficient information to model rare subgroups.

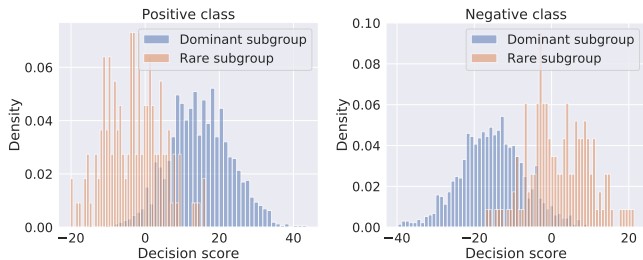 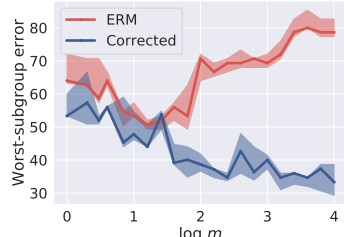

(a) Distribution of overparameterised model scores.     (b) Effect of post-hoc correction.

Figure 1: Distribution of model decision scores on test samples from a synthetic dataset of Sagawa et al. (2020b), comprising two labels with two subgroups each (left panel). The scores are expected to be $> 0$ iff the label is positive. We train a linear model with complexity controlled by its number of features $m$. For the overparameterised setting $m = 10^4$ (left), within each class, rare subgroups consistently appear on the wrong side of the decision boundary. Correcting this bias via post-hoc score translation improves the worst-subgroup error as the model complexity is increased (right).

To make the above concrete, Figure 1 plots a histogram of model predictions for a synthetic dataset from Sagawa et al. (2020b) (cf. §2). The data comprises four subgroups generated from combinations $(y, a(x))$ of labels $y \in \{\pm 1\}$ and a feature $a(x) \in \{\pm 1\}$. Most samples $(x, y)$ have $y = a(x)$, and so these comprise two dominant subgroups within the positive and negative samples. We train an overparameterised linear model, yielding logits $f_{\pm 1}(x)$. We then plot the decision scores $f_{+1}(x) - f_{-1}(x)$, which are expected to be $> 0$ iff $y = +1$. Strikingly, there is a distinct separation amongst the subgroup scores: e.g., samples with $y = +1, a(x) = -1$ have systematically lower scores than those with $y = +1, a(x) = +1$. Consequently, the model incurs a significant error rate on rare subgroups. The structured nature of the separation implies suggests to post-hoc shift the scores to align the distributions; this markedly improves performance on the rare subgroups (Figure 1b).

**Scope and contributions**. The primary aim of this work is furthering the understanding of the behaviour of overparameterised models, rather than proposing new techniques. Indeed, the post-hoc correction techniques we employ have been well-studied in the related problem setting of *long-tail learning* or learning under *class imbalance* (He & Garcia, 2009; Buda et al., 2017; Van Horn & Perona, 2017). Several works have demonstrated that the representations learned by standard networks contain sufficient information to distinguish between dominant and rare labels (Liu et al., 2019; Zhang et al., 2019; Kang et al., 2020; Menon et al., 2020). Similar techniques are also common the fairness literature (Hardt et al., 2016; Chzhen et al., 2019). However, it is not *a-priori* clear whether such techniques are effective for overparameterised models, whose ability to perfectly fit the training labels can thwart otherwise effective approaches (Sagawa et al., 2020a).

Existing techniques for improving the worst-subgroup error of overparameterised models involve altering the inputs to the model (Sagawa et al., 2020b), or the training objective (Sagawa et al., 2020a). By contrast, the techniques we study alter the *outputs* of a standard network, trained to minimise the softmax cross-entropy on the entire data. Our findings illustrate that such models do not *necessarily* require bespoke training modifications to perform well on rare subgroups: even with standard training, overparameterised models can (in certain settings) learn useful information about rare subgroups.

In summary, our contributions are:

(i) we demonstrate that, in certain settings, overparameterised models' poor performance on under-represented subgroups is the result of a structured bias in the classification layer (cf. §3);

(ii) we show that two simple post-hoc correction procedures (cf. §4) can mitigate the above bias, and thus significantly reduce their worst-subgroup error (cf. §5).

## 2   BACKGROUND AND SETTING

Suppose we have a labelled training sample $\mathcal{S} = \{(x_i, y_i)\}_{i=1}^n \in (\mathcal{X} \times \mathcal{Y})^n$, for instance space $\mathcal{X} \subset \mathbb{R}^d$ and label space $\mathcal{Y}$. One typically assumes $\mathcal{S}$ is an i.i.d. draw from some unknown distribution $\mathbb{P}(x, y)$. Further, suppose each $(x, y)$ has an associated *group membership* $g(x, y) \in \mathcal{G}$, with

$G \doteq |\mathcal{G}|$. This induces $G$ data *subgroups*, with a prior $\mathbb{P}(g)$ and conditional distributions $\mathbb{P}(x, y \mid g)$. Following Sagawa et al. (2020a;b), we consider groups $g(x, y) = (y, a(x))$, where $a(x) \in \mathbb{R}$ is some attribute within $x$. We assume $a(x)$ is fully specified during train and test time; while not always realistic, such an assumption has precedent in the fairness literature (Lipton et al., 2018).

The standard goal in classification is to learn a classifier $h \colon \mathcal{X} \to \mathcal{Y}$ that minimises the *average error*

$$L_{\mathrm{avg}}(h) \doteq \mathbb{E}_{g} \mathbb{E}_{x, y | g} \left[ \ell_{01}(y, h(x)) \right],$$

where $\ell_{01}(y, h(x)) = [\![ y \neq h(x) ]\!]$ is the 0-1 loss. Typically, one constructs $h(x) = \operatorname{argmax}_y f_y(x)$, where $f(x) \in \mathbb{R}^{\mathcal{Y}}$ comprises real-valued *logits*, as learned by *empirical risk* minimisation (ERM): $\min_{f \in \mathcal{F}} \frac{1}{n} \sum_{i=1}^{n} \ell(y_i, f(x_i))$. Here, $\ell$ is a surrogate loss such as the softmax cross-entropy, and $\mathcal{F}$ is a function class, such as a neural networks with a fixed architecture. A network is *overparametrised* if it can *perfectly* fit the training labels, and thus drive the training error to zero. Remarkably — and in apparent contrast to orthodox statistical wisdom — this does not come at the expense of generalisation on test samples (Zhang et al., 2017; Belkin et al., 2019; Nakkiran et al., 2020).

This apparent power comes at a price, however. Let us define the *worst-subgroup error* as

$$L_{\mathrm{max}}(h) \doteq \max_{g \in \mathcal{G}} \mathbb{E}_{x, y | g} \left[ \ell_{01}(y, h(x)) \right], \tag{1}$$

i.e., the worst-case error over all data subgroups. Prior work (Sagawa et al., 2020a;b) established that for overparameterised models, the worst-subgroup *training* error can go to zero (since the model can fit all samples), but the worst-subgroup *test* error can devolve to that of random guessing (since the model can fit spurious correlations for rare subgroups). Further, the degree of degradation can increase with the model complexity. This indicates that the naïve use of overparametrised models may be at odds with ensuring *fairness* across data subgroups, a core concern in moden applications of machine learning (Calders & Verwer, 2010; Dwork et al., 2012; Hardt et al., 2016; Zafar et al., 2017).

There are several potential strategies to cope with this. One is to perform distributionally robust optimisation (Hashimoto et al., 2018; Mohri et al., 2019; Sagawa et al., 2020a), and minimise:

$$L_{\mathrm{DRO}}(h) \doteq \max_{g \in \mathcal{G}} \left[ \mathbb{E}_{x, y | g} \left[ \ell(y, f(x)) \right] + \Omega_g(f) \right],$$

where $\Omega_g$ is some per-group regulariser. In settings where $\mathbb{P}(g)$ is non-uniform, Sagawa et al. (2020a) proposed to set $\Omega_g(f) \equiv \frac{1}{\sqrt{n_g}}$, where $n_g$ is the number of training samples with group $g$. Alternatively, one can reweight samples to upweight the contribution of rarer groups and minimise:

$$L_{\mathrm{RW}}(h) \doteq \sum_{g \in \mathcal{G}} w_g \cdot \mathbb{E}_{x, y | g} \left[ \ell(y, f(x)) \right], \tag{2}$$

where, e.g., $w_g = \mathbb{P}(g)$ leads to the standard average error, while $w_g = 1$ implicitly upweights rare subgroups. While intuitive, Sagawa et al. (2020b) established that such an approach is also subject to poor worst-subgroup performance, owing to a broader issue with using importance weighting in conjunction with neural networks (Wen et al., 2014; Byrd & Lipton, 2019). Sagawa et al. (2020b) established that one can achieve good performance by instead *subsampling* dominant groups, an operation equivalent in expectation to minimising $L_{\mathrm{RW}}(h)$ with $w_g = 1$. Recent developments in the mitigation of worst-subgroup errors include Nam et al. (2020); Zhang et al. (2020); Goel et al. (2020).

In the sequel, we shall make extensive use of three datasets from Sagawa et al. (2020a;b), each of which involve binary labels $y \in \mathcal{Y}$ and a binary attribute $a(x) \in \mathcal{A}$:

(i) `synth`, a synthetic dataset where $\mathcal{X} \subset \mathbb{R}^{200}$, $\mathcal{Y} = \{\pm 1\}$, and $\mathcal{A} \in \{\pm 1\}$.

(ii) `waterbirds`, a dataset of bird images with $\mathcal{Y} = \{\texttt{land bird}, \texttt{water bird}\}$ corresponding to the bird type, and $\mathcal{A} = \{\texttt{land background}, \texttt{water background}\}$ corresponding to the background.

(iii) `celebA`, a dataset of celebrity images with $\mathcal{Y} = \{\texttt{blond}, \texttt{dark}\}$ corresponding to individuals' hair colour, and $\mathcal{A} = \{\texttt{male}, \texttt{female}\}$.

For each dataset, we construct four subgroups $g(x, y) = (y, a(x))$, with two such subgroups being under-represented. On `synth` and `waterbirds`, these correspond to subgroups with $y \neq a(x)$, while on `celebA`, these corespond to the subgroups $\{(\texttt{blond}, \texttt{male})\}$ and $\{(\texttt{dark}, \texttt{female})\}$. Owing to

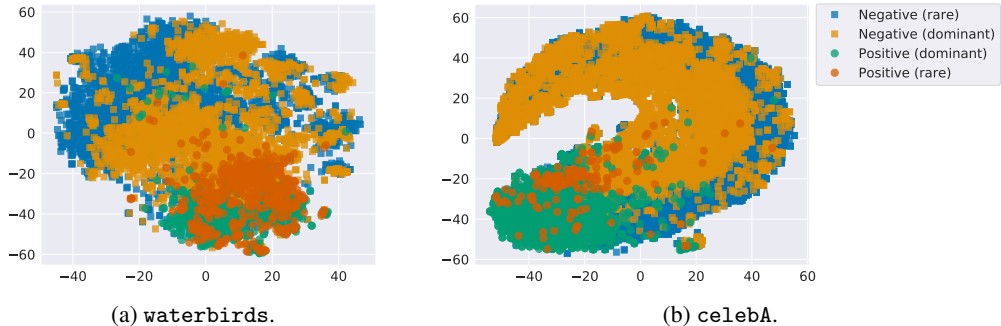

(a) `waterbirds`.  (b) `celebA`.

Figure 2: Two-dimensional tSNE visualisation of test embeddings as produced by overparameterised models. Within each class, samples from each of the two subgroups tend to be closely clustered. This suggests that the *representations* learned by the models contain sufficient information to help distinguish samples from rare versus dominant subgroups. The poor worst-subgroup performance of such models may thus be traced to issues with the *classification* layer. (Best viewed in colour.)

the rarity of certain subgroups, it is intuitively easy for an overparameterised network to learn to predict $a(x)$ rather than $y$, and memorise spurious patterns to predict the rare subgroups.

To train overparameterised models, we follow the setup of Sagawa et al. (2020a;b), which we briefly summarise. For `celebA` and `waterbirds`, we use a ResNet-50, which can attain perfect training accuracy. For `synth`, we train a weakly regularised ($\lambda = 10^{-16}$) logistic regression model on a *fixed* representation $\Phi$ constructed as follows: for fixed $m$, we construct $\Phi(x) = \text{ReLU}(Vx)$, where $V \in \mathbb{R}^{m \times 200}$ is a random Gaussian matrix with normalised rows. Overparameterised models consistently demonstrate a significant gap between the average and worst-subgroup error: e.g. (see Figure 4), on `synth`, the model achieves 91% average accuracy, but 36% worst-subgroup accuracy.

## 3   THE DOMINANT SUBGROUP BIAS OF OVERPARAMETERISED MODELS

We study the nature of overparameterised models' poor performance on rare subgroups more closely. We make two observations: first, this under-performance is largely owing to a bias in the *classification* layer. Second, this bias manifests in the form of a *distribution shift* in the model scores for rare subgroups. This shall subsequently motivate post-hoc correction procedures.

### 3.1   CAN THE LEARNED REPRESENTATIONS DISTINGUISH BETWEEN SUBGROUPS?

Neural models make predictions based on logits $f_y(x) = w_y^\top \Phi(x)$, for classification weights $w_y \in \mathbb{R}^K$ and representations $\Phi(x) \in \mathbb{R}^K$. Suppose such a model performs poorly on a subgroup $(\bar{y}, \bar{a})$. This implies that for a sample $(x, \bar{y})$ in this subgroup, $f_{\bar{y}}(x) \ll f_{y'}(x)$ for some competing $y'$.

Why do overparameterised models underperform on rare subgroups? The factorised nature of the logits suggests this arises from issues with the representations, the classification weights, or both. To determine which of these is likely, we begin by inspecting the representations. Recall that we train a ResNet-50 on `celebA` and `waterbirds`, which produce $K = 2048$ dimensional instance embeddings. We may thus embed *test* instances from each of the four data subgroups, and study their geometric structure. Intuitively, if instances from rare and dominant subgroups with the same label have limited similarity, the representations for rare samples are insufficiently rich.

The high dimensionality of the embeddings prohibits an exact inspection, but as a rough surrogate, we employ a two-dimensional tSNE (Maaten & Hinton, 2008) visualisation. As tSNE attempts to preserve neighbourhood information amongst samples, we hope to evince the relative geometries of samples from the various subgroups. Figure 2 reveals that for `celebA` and `waterbirds`, samples from the rare subgroups tend to be closely clustered with those belonging to the same class.

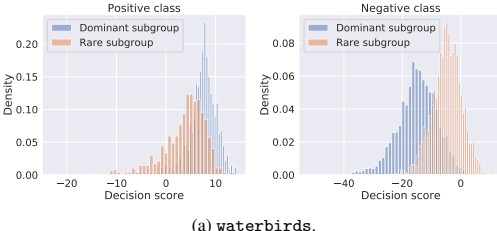 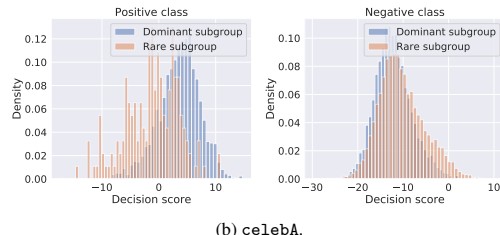

(a) `waterbirds`.              (b) `celebA`.

Figure 3: Distributions of model decision scores $f_{+1}(x) - f_{-1}(x)$ on test samples on `waterbirds` and `celebA`, comprising two labels with two sub-groups each. The rare and dominant subgroup score distributions largely differ, with the former often on the incorrect side of the decision boundary (i.e., with decision scores $< 0$ for positive samples, and $> 0$ for negative samples). Notably, the scores for rare subgroups tend to be shifted compared to those for dominant subgroups.

The above suggests that the *representations* learned by the models contain information to (at least partly) help distinguish samples from rare versus dominant subgroups.[1] This suggests the poor worst-subgroup performance of such models may result from issues with the *classification* layer.

### 3.2 HOW DOES BIAS MANIFEST IN THE CLASSIFICATION LAYER?

To study the potential issues in the classification layer, we continue our strategy of visualising the model outputs. Since the datasets we consider involve binary labels, we may simply study the distribution of the decision scores $f_{+1}(x) - f_{-1}(x)$. Since our predicted label is the highest scoring logit, we desire this score to be $> 0$ iff the sample has a positive label. As in the previous section, we may break down these scores for each of the subgroups induced by label $y$ and attribute $a(x)$.

We earlier illustrated this distribution for `synth` in Figure 1; Figure 3 further provides distributions for `celebA` and `waterbirds`. On both datasets, within each label, there is a *shift* in the score distributions for one or both of the rare subgroups. This is in keeping with the poor performance of the model on these subgroups: e.g., on `waterbirds`, rare samples in the positive class systematically have *negative* decision scores, implying the model incurs a high false-negative rate on these samples.

In light of the above findings, we may revisit ablations from prior work that tease apart the factors causing poor worst-subgroup performance. For example, Sagawa et al. (2020b) showed that increasing model complexity on `synth` can degrade worst-subgroup performance beyond a certain critical point, and that this can be mitigated with a combination of strong regularisation and subsampling. The same conclusions largely hold (see Appendix D) for the varying decision scores amongst subgroups: e.g., the difference in rare and dominant subgroup scores is exacerbated as we increase model complexity.

### 3.3 DISCUSSION AND IMPLICATIONS

The systematic under-prediction of scores for rare subgroups can be seen as a particular manifestation of neural networks producing uncalibrated probability estimates (Guo et al., 2017): from Figure 3, the model will systematically under- or over-estimate the probability of rare samples being positive.

We emphasise here that our illustrations above are for *test* samples not observed during training. Training samples exhibit qualitatively different trends, reflective of overparameterised models' ability to perfectly fit them: e.g., the decision scores for all samples are consistently on the correct side of the decision boundary (see Appendix D). The fact that the scores on unseen samples exhibit a distinction amongst subgroups suggest the network encodes an implicit bias against such samples.

At the same time, this bias largely manifests as a *translation* of the scores. This suggests a simple post-hoc correction of the scores may suffice to improve performance; e.g., bumping up scores for samples with $a(x) =$ `land background` in `waterbirds` can make the error on the rare subgroup (`water bird`, `land background`) more equitable. It remains now to more carefully describe such post-hoc procedures, and study their performance.

---

[1]We crucially rely on a classification layer to help distinguish samples between subgroups; by themselves, however, the embeddings can be systematically biased (Bolukbasi et al., 2016; Gonen & Goldberg, 2019).

## 4 CORRECTING THE SUBGROUP BIAS OF OVERPARAMETERISED MODELS

Drawing from the literature on long-tail learning (Zhang et al., 2017; Kang et al., 2020; Ye et al., 2020) and fairness (Hardt et al., 2016; Chzhen et al., 2019), we now detail two post-hoc correction techniques to mitigate overparameterised models' bias against rare subgroups.

### 4.1 CLASSIFIER RETRAINING

Given that §3.1 demonstrates that learned representations $\Phi(x)$ appear meaningful across subgroups, a natural thought is to fit a linear classifier on top of them; i.e., we treat $\{(\Phi(x_i), y_i)\}_{i=1}^{n}$ as a new training set for a linear model. Overparameterisation introduces a challenge, however: since the original network can find a classifier with the lowest possible (i.e., zero) training error, simple modifications to the loss (e.g., reweighting samples) will result in learning the same classifier.

Fortunately, there are several options to find a distinct classifier than the original network: e.g., one can subsample elements from the majority subgroups, per Sagawa et al. (2020b). We emphasise an important difference between employing such techniques in standard training, and in classifier retraining: the latter uses representations learned from a standard network, as opposed to changing the network objective itself. The success of the latter shall thus demonstrate that the standard network representations are rich enough to reasonably distinguish between subgroups.

### 4.2 THRESHOLD CORRECTION

The illustrations in Figures 3 suggest a simple approach to improving classification performance: rather than using an identical classification threshold for all samples, one can employ per-subgroup thresholds. In detail, observe that in the case of binary labels, we predict $h(x) = +1 \iff f_{+1}(x) - f_{-1}(x) > 0$. Instead, we can predict

$$h(x) = +1 \iff f_{+1}(x) - f_{-1}(x) > t_{a(x)},$$

where $\{t_a \in \mathbb{R} : a \in \mathcal{A}\}$ are per-attribute thresholds. Equivalently, this translates the scores for all samples with a given attribute $a(x)$ so that the decision boundary is at 0. This can compensate for the distribution shifts observed in Figures 1 and 3: intuitively, by enforcing a lower threshold for samples with $a(x) = -1$, we can account for the fact that most of these samples obtain a low model score.

It remains to specify how to choose the thresholds $t_a$. One simple option is to perform a parameter sweep, and employ the thresholds that minimise the worst-subgroup error on a holdout set. This is feasible in settings where $|\mathcal{A}|$ and $|\mathcal{Y}|$ are small, and can directly target the performance measure of interest. For more complex problems, one may suitably parameterise the thresholds, or attempt to *learn* so as to minimise a suitable objective (see discussion below).

### 4.3 DISCUSSION AND RELATED WORK

The long-tail learning literature has demonstrated the value of both classifier retraining (Zhang et al., 2017; Kang et al., 2020) and threshold correction (Zhou & Liu, 2006; Collell et al., 2016; Menon et al., 2020). Similarly, in the fairness literature, post processing of classifier outputs based on per-subgroup thresholds is a similarly well-established technique (Hardt et al., 2016). Our aim here is to investigate the effectiveness of such techniques in the overparameterised setting. Given such models can perfectly fit training labels, it is less clear whether the representations learned from standard training are sufficiently useful to learn a good classifier, and whether their outputs can be easily corrected post-hoc; e.g., a model that merely memorised certain training labels could not be meaningfully corrected to perform well on test samples.

To extend threshold correction to multi-class settings, one could tie the thresholds to the frequencies $\mathbb{P}((y, a(x)))$ of various subgroups, rather than tune them. This is akin to class prior or *logit correction* techniques from long-tail learning (Collell et al., 2016; Menon et al., 2020). In the fairness literature, relevant techniques include Hardt et al. (2016), who propose an objective to select optimal thresholds for ensuring a particular notion of fairness; and recent techniques that learn a post-processing of model scores (Chzhen et al., 2019; Jiang et al., 2020; Wei et al., 2020) to de-bias predictions. Exploring such techniques in the overparameterised setting is an interesting direction for future work.

Table 1: Summary of average and worst-subgroup errors on `synth`, `waterbirds`, and `celebA`. For `synth` and `waterbirds` with a logistic regression model, we use the overparameterised setting where $m = 10^4$. Standard ERM performs well on average, but poorly on the worst-subgroup. However, this is significantly improved by post-hoc correction of the ERM predictions, either through classifier retraining (CRT) or threshold correction (THR). The former is only applied to models with a learned representation. We also report the results of distributionally robust optimisation (DRO) and sub-sampling of the training data (SAM), per Sagawa et al. (2020a;b). The results marked with † are quoted from Sagawa et al. (2020a, Table 2).

| Dataset | Model | Average error | | | | | Worst-subgroup error | | | | |
|---|---|---|---|---|---|---|---|---|---|---|---|
| | | ERM | SAM | DRO | CRT | THR | ERM | SAM | DRO | CRT | THR |
| `synth` | Logistic | 9.49 | 23.99 | 17.99 | N/A | 22.18 | 63.20 | 27.38 | 22.67 | N/A | 27.51 |
| `waterbirds` | Logistic | 4.37 | 17.18 | 10.57 | N/A | 16.71 | 52.34 | 20.82 | 16.73 | N/A | 17.46 |
| | ResNet | 2.15 | 15.66 | $6.30^{\dagger}$ | 2.58 | 5.23 | 17.23 | 18.41 | $9.50^{\dagger}$ | 8.30 | 8.70 |
| `celebA` | ResNet | 4.64 | 9.28 | $6.60^{\dagger}$ | 9.38 | 7.90 | 56.11 | 10.63 | $12.20^{\dagger}$ | 14.68 | 12.96 |

The contemporaneous work of Sohoni et al. (2020) also considers the viability of improving worst-subgroup performance using the learned representations. Specifically, they propose to cluster these embeddings, and train a new distributionally robust classifier to predict cluster assignments. While very much in the spirit of our classifier retraining proposal, their study does not consider the efficacy of such a procedure under varying model complexity, nor correction techniques that operate purely on the classification outputs.

## 5 EXPERIMENTS: HOW EFFECTIVE IS POST-HOC CORRECTION?

We now show that the above post-hoc correction techniques can significantly improve overparameterised models' performance on rare subgroups. This indicates that even when trained in the usual manner, such models can learn useful information about rare subgroups.

### 5.1 EXPERIMENTAL SETUP

We follow the same basic setup as Sagawa et al. (2020a;b). We instantiate overparameterised models on each dataset: a ResNet-50 on `celebA` and `waterbirds`; a linear logistic regression on `synth`, using fixed features as described in §2; and a linear logistic regression on `waterbirds`, using the embeddings from a ResNet-18 pre-trained on ImageNet. See Appendix A for details of experimental hyper-parameters. We measure both the *average* and *worst-subgroup* errors on both the train and test set, repeating each experiment 5 times.

We apply post-hoc correction to these learned models, via classifier retraining (**CRT**) on the learned representations, using a linear logistic regression model with subsampling of the dominant subgroups per Sagawa et al. (2020b); and threshold correction (**THR**) on the decision scores, using a holdout set to estimate thresholds $\{t_a : a \in \{\pm 1\}\}$ that minimise the worst-subgroup error. For `waterbirds`, we use the holdout set from Sagawa et al. (2020a); for `celebA`, we use the standard holdout set; and for `synth`, we construct a holdout set using 20% of the training samples.

As reference, we report the results of standard minimisation on a *balanced subsample* (**SAM**) of the training set, where following Sagawa et al. (2020b) we down-sample each subgroup to have $n_{\min}$ examples, where $n_{\min}$ is the number of examples in the smallest subgroup. We also report the results of the regularised distributionally robust optimisation (**DRO**) procedure of Sagawa et al. (2020a). This involves strongly regularising the model, and modifying the training objective to target the worst-subgroup error. Our aim is *not* to improve on the performance of this method, which directly trains to minimise the worst-subgroup error; rather, our goal is to understand and quantify how much useful information standard model training learns about rare subgroups.

### 5.2 RESULTS AND DISCUSSION

Table 1 summarises the test set results on all datasets. We highlight several key points.

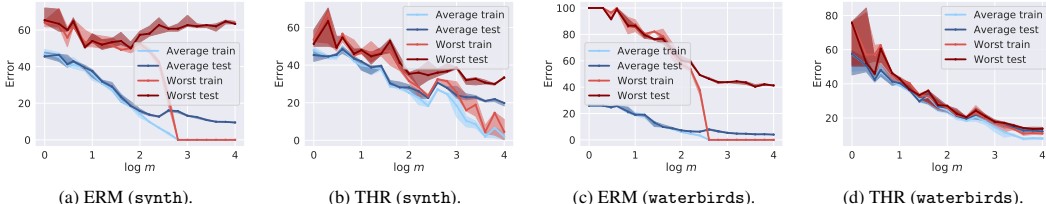

|  |  |  |  |
|---|---|---|---|
| (a) ERM (synth). | (b) THR (synth). | (c) ERM (waterbirds). | (d) THR (waterbirds). |

Figure 4: Performance of baseline (ERM) and threshold correction (THR) on synth and waterbirds datasets, with fixed input representations. We vary the feature dimensionality $m$ between $10^1$ and $10^4$, and plot the average and worst-subgroup error on both the train and test set over 5 independent trials. For the baseline, the worst-case error devolves as $m$ increases. For threshold correction, both average and worst-case error steadily improve on the test set. Of salience is that overparameterisation (large $m$) aids worst-subgroup performance under post-hoc correction.

**ERM has poor worst-subgroup error**. In keeping with Sagawa et al. (2020a;b), standard ERM performs poorly in terms of worst-subgroup error. This may be strongly mitigated by modifying the training objective per DRO, or by modifying the training sample per SAM. (Each of these incurs only a mild penalty in terms of the average error.) This confirms the finding of prior work that modification of the training procedure can achieve a suitable trade-off between average- and worst-subgroup error.

**Post-hoc correction improves worst-subgroup error**. Encouragingly, post-hoc techniques consistently and significantly reduce the worst-subgroup error of ERM. For example, on celebA, THR reduces the worst-subgroup error from 56.11% to 12.96%, with only a mild increase in the average error. Between the post-hoc techniques, we generally find THR to yield the best performance. We reiterate that this technique does not modify the training procedure directly, but simply post-processes the learned ERM model output. This confirms the analysis of the previous section, which indicates that (on the considered datasets) the result of standard ERM can by themselves contain sufficient information to overcome poor worst-subgroup error.

**Post-hoc correction is comparable to training modification**. Post-hoc techniques generally compare favourably with DRO and SAM in terms of the trade-off between average- and worst-subgroup error. While DRO is notably superior on synth, this gap appears to be in part due to the challenge of tuning hyperparameters given holdout data with limited subgroup representation. For example, using a larger holdout set size of $n = 2400$ samples improves the performance of THR on synth to $24\%$.

The superior overall performance of DRO is in keeping with findings about the efficacy of training modification for fairness (Agarwal et al., 2018). Nonetheless, the generally competitive performance of CRT and THR suggests the latter can extract non-trivial gains from overparameterised models.

**Overparameterisation with post-hoc correction can improve worst-subgroup error**. We now confirm that increasing model complexity can result in improved average *and* worst-subgroup performance, *provided* the ERM outputs are suitably corrected. As the synth and waterbirds datasets with fixed features allow for modifying the feature dimensionality $m$ — which controls the degree of overparameterisation – we perform an experiment where we vary $m$ between $10^1$ and $10^4$. Figure 4 shows the average and worst-subgroup error on both the train and test set, over 5 independent trials. The overall and worst-case training error approaches 0% as $m$ increases (i.e., the model becomes overparameterised). Further, the average test error appears reasonable, approaching 10%; however, the worst-case error devolves as $m$ increases, exceeding 60%.

The plots also show the results of THR as $m$ is varied. Here, both average and worst-case error steadily improve: indeed, overparameterisation *aids* worst-subgroup performance. Further, the gains in the worst-subgroup test error are dramatic compared to ERM. Thus, while ERM is susceptible to a tradeoff between overparameterisation and worst group accuracy, this tension can be mitigated with simple mechanisms to encourage equitable performance. This general trend is largely robust to distributional properties such as the fraction of majority samples in synth, though significantly increasing this expectedly degrades the best achievable performance; see Appendix D.

**Illustration of classification thresholds**. Figure 5 studies the effect of tuning the classification thresholds $t_a$ on each of the per-subgroup errors for synth ($m = 10^4$). Here, we plot the errors for

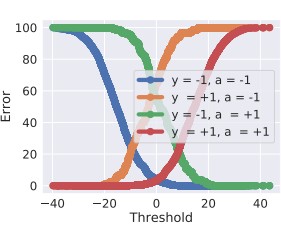
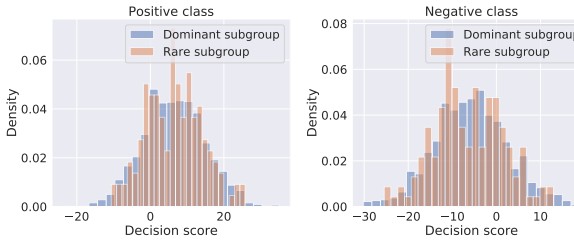

|(a) Effect of threshold $t_a$.|(b) Distribution of scores with optimal threshold.|

Figure 5: Effect of tuning classification threshold $t_a$ for `synth`, when $m = 10^4$. The case $t_a = 0$ corresponds to the baseline. Modifying $t_a$ systematically trades off performance on the rare subgroups compared to the dominant ones (left). By choosing the thresholds to minimise the worst-subgroup error, the resulting score distributions are aligned for rare and dominant subgroups (right).

each of the four subgroups as we sweep over the thresholds. The case $t_a = 0$ corresponds to the baseline. Modifying $t_a$ trades off performance on the rare subgroups compared to the dominant ones. One may then pick optimal thresholds for $a = \pm 1$, which correspond to cross-over points of the error curves for each value of $a$: for example, when $t_{+1} \sim 10$, the errors on the subgroups $y = +1, a = +1$ and $y = -1, a = +1$ are equitable, and thus the maximum of the two errors is minimised. Choosing such thresholds aligns the scores distributions for rare and dominant subgroups (right).

## 6 DISCUSSION AND FUTURE WORK

Post-hoc correction relies on knowledge of the data subgroups at train and test time. An important practical challenge is extending this to settings with *unknown* subgroups. One natural strategy is to attempt to unearth these subgroups via clustering of the model outputs, but careful study is needed to inform design choices (e.g., choosing the number of clusters). For a contemporaneous study of a similar technique, see Sohoni et al. (2020). Exploring the viability of post-hoc correction in overparameterised settings with multi-class labels, and multiple subgroups — e.g., through adaptations of techniques noted in §4.3 — is also of interest. More broadly, the study of learning bias-free representations has received significant interest (Bahng et al., 2020; Kim et al., 2019; Li & Vasconcelos, 2019; Arjovsky et al., 2020; Nam et al., 2020). Exploring the efficacy of such approaches in overparameterised settings would be of interest.

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

## A    EXPERIMENTAL SETUP

For the logistic regression experiments, we train the models using `LogisticRegression` package in `sklearn` with $C = 1/(n \cdot \lambda)$, for number of training examples $n$ and minimal regularisation strength $\lambda = 10^{-16}$. This employs the `liblinear` solver under-the-hood, and thus finds an approximate minimiser for the convex empirical risk objective on the entire training sample.

For the ResNet-50 experiments, we initalise the model using a ResNet-50 pre-trained on ImageNet. We train the models using SGD with a momentum value of $0.9$. We use a batch size of $128$, weight decay $10^{-4}$, and a learning rate of decayed according to a cosine schedule. We train with a base learning rate of $10^{-4}$ for $1000$ epochs[2] on `waterbirds`, and a base learning rate of $10^{-2}$ for $50$ epochs on `celebA`. We also employ data augmentation in the form of random cropping and flipping, which offers a consistent boost in rare subgroup performance.

## B    ADDITIONAL EXPERIMENTS

We present additional experimental results that explore a few key hypotheses:

- how sensitive are the results to inexact subgroup specification?
- how does the precise choice of target label and spurious attribute affect results?
- how generalisable are the results to multi-class settings?

### B.1    INEXACT SUBGROUP SPECIFICATION

The post-hoc modification techniques in the body crucially rely on knowledge of the precise subgroup specification of each example. This is unrealistic in practice, where the subgroups may be latent or inexactly specified. Following Sagawa et al. (2020a)[Appendix B], we simulate a setting of inexact specification of the subgroups on `celebA`. Here, we use as spurious attributes $\mathcal{A}' = \texttt{WearingLipstick} \times \texttt{Eyeglasses} \times \texttt{Smiling} \times \texttt{DoubleChin} \times \texttt{OvalFace}$, comprising 32 distinct values. We then learn using the subgroups $\mathcal{Y} \times \mathcal{A}'$ as input, and then measure performance with respect to the *original* subgroups $\mathcal{Y} \times \mathcal{A}'$.

As noted in the body, the threshold adjustment technique (THR) is challenging to apply as-is in this setting, as it requires setting 32 distinct thresholds. As suggested in §4.3, we thus apply a simple heuristic of tying the thresholds to the subgroup frequencies, i.e., $t_{a(x)} = \log \mathbb{P}(y = +1 \mid a(x)) - \log \mathbb{P}(y = -1 \mid a(x))$. This has the effect of implicitly requiring higher model confidences to classify examples into a dominant subgroup. This technique is seen to have a worst-subgroup error of $16.67\%$, which is a modest increase compared to the $12.10\%$ obtained when using the exact subgroups $\mathcal{Y} \times \mathcal{A}'$. This illustrates that one can still make useful predictions given imperfect subgroup information.

### B.2    CHOICE OF TARGET LABEL AND SPURIOUS ATTRIBUTE

The results in the body involve data with one or more rare subgroups. Is this rarity the primary factor that influences performance, or does the definition of the subgroups themselves matter? To test this, we consider a variant of the `celebA` dataset in the body where the target label and sensitive attribute are swapped. In this variant of the dataset, we have $\mathcal{Y} = \{\texttt{male}, \texttt{female}\}$ and $\mathcal{A} = \{\texttt{blond}, \texttt{dark}\}$. This exactly preserves the subgroup definitions and their rarity, but fundamentally changes the target label and feature used in training.

Interestingly, this simple modification dramatically improves performance of the baseline: on the rarest subgroup, the error is $13.67\%$, which is a significant reduction over the $56.94\%$ for the original dataset. This indicates that the precise choice of subgroup definition can play a non-trivial role in final performance. Intuitively, performance can be hampered when the target variable is spuriously correlated with many features in the training set.

---

[2]This large number of epochs, coupled with the small learning rate, endows some stability in the worst subgroup performance. One may however obtain qualitatively similar results with much fewer epochs and a larger learning rate.

Nonetheless, even with this improved model, we find that threshold adjustment (THR) can further improve performance to $9.11\%$. The average subgroup errors of both techniques are similar, being $1.29\%$ and $1.28\%$ respectively.

### B.3    MULTI-CLASS SETTINGS

The results in the body involve problems with binary labels. To assess the effect of working with multi-class labels, we employ a modified version of MNIST based on Goel et al. (2020). Here, one mixes the standard MNIST dataset with samples from a corrupted version of MNIST comprising zig-zag images. The zig-zag images are made to be strongly correlated with the digit parity, so that most odd digits are zig-zagged. We then consider subgroups defined by $\mathcal{Y} \times \mathcal{A}$, where $\mathcal{A} = \{\texttt{normal}, \texttt{zig} - \texttt{zag}\}$. Note that we consider $\mathcal{Y} = \{0, 1, \ldots, 9\}$ to illustrate performance in a multi-class setting, unlike Goel et al. (2020) who consider $\mathcal{Y} = \{0, 1\}$ to be the digit parity.

We train a LeNet-5 for 100 epochs using a learning rate of 0.0001, momentum 0.9, weight decay 0.05, and batch size 100. Here, ERM achieves a worst-subgroup error of $67.11\%$. Classifier retraining (cRT) based on subsampling all non-minority samples improves this to $74.52\%$. Similarly, threshold adjustment (THR) based on the heuristic of tying the thresholds to the subgroup frequencies (as described in the previous section) achieves $78.95\%$. This illustrates the potential for post-hoc techniques to also be useful in scenarios other than binary classification.

## C    VISUALISATION OF EMBEDDINGS UNDER GROUP-BASED DRO

Figure 6 shows a tSNE visualisation of the embeddings learned by a model trained to minimise the group-based DRO objective of Sagawa et al. (2020a). Similar to the results of ERM, there is generally a notable separation of samples from the four subgroups.

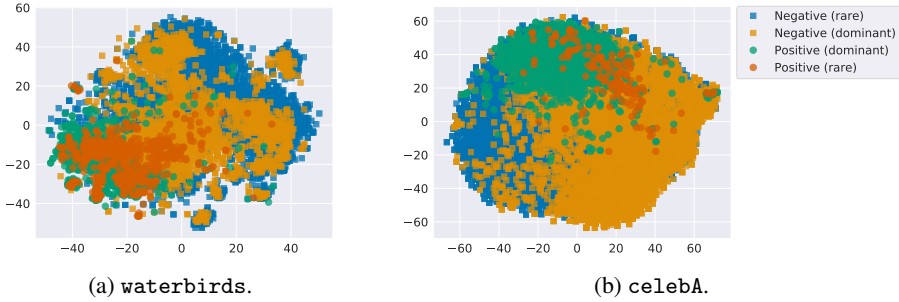

(a) `waterbirds`.                    (b) `celebA`.

Figure 6: Two-dimensional tSNE visualisation of test embeddings as produced by an overparameterised model trained to minimise group-based DRO.

## D    ADDITIONAL EXPERIMENTAL ABLATIONS

We present additional experimental results, highlighting several key points:

- the separation of scores between rare and dominant subgroups is consistent across all datasets considered in the paper; however, the score distributions are markedly different on train and test sets, owing to models being overly confident on training samples.
- increasing model complexity systematically exacerbates the distribution shift in decision scores.
- early stopping has limited effect on the score distributions; even after a single of epoch of training, there may be a distinction between rare and dominant subgroups' scores.
- increased $\ell_2$ regularisation strength has a favourable effect on the score distributions, encouraging samples from both rare and dominant subgroups to be correctly classified.
- subsampling (per Sagawa et al. (2020a)) has a positive effect on the score distributions, making them almost perfectly align across subgroups.

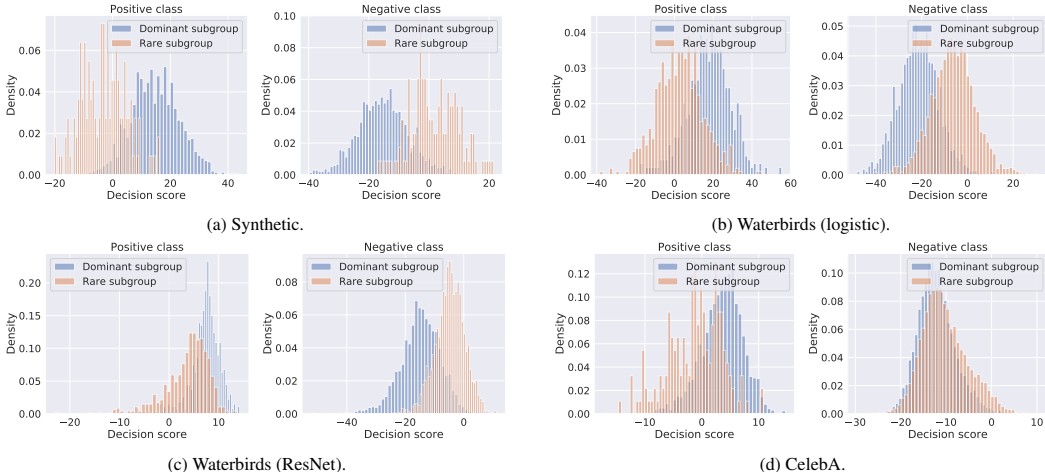

Figure 7: Histograms of model scores on test samples on various datasets, comprising two labels with two sub-groups each. In general, there are differences in the distributions for the rare and dominant subgroup scores, with the former often lying on the incorrect side of the decision boundary.

- increasing the fraction of majority samples has a deleterious effect on overall performance; however, even at extreme levels of imbalance, the score distribution for rare samples may be shifted to correct for bias.

### D.1 HISTOGRAM OF TRAIN AND TEST SCORES

Figure 7 plots histograms of test scores for all datasets considered in this paper. We consistently find that there is a separation between the scores for rare and dominant subgroups.

We see similar behaviour on training scores in Figure 8. However, note the vastly different scale, owing to the model being more confident in its predictions for these samples. In general, while there are differences in the distributions for the rare and dominant subgroup scores, nearly all such scores are on the correct side of the decision boundary. This is expected, since overparameterised models perfectly fit the training data, and thus correctly classify all samples. The ability of these models to nonetheless produce meaningful results on test samples is owing to their inductive bias.

### D.2 IMPACT OF MODEL COMPLEXITY ON SCORES

Figure 9 shows model scores on test samples on `synth` as number of projection features $m$ is varied. We see that as the model complexity increases, there is a steady increase in the separation of decision scores between the rare and dominant subgroups for a label. This is in keeping with overparameterisation exacerbating worst-subgroup error: as the decision scores have more pronounced separation, using a default classification threshold will lead to significantly worse performance.

### D.3 IMPACT OF EARLY STOPPING ON SCORES

Figures 10 and 11 shows the evolution of model scores on test samples on the CelebA and Waterbirds datasets. Here, the distinction between the scores amongst subgroups of the positive class is visible even after early stopping. With increased training epochs, there is a systematic shift of the negative scores, as the network becomes increasingly confident on them.

### D.4 IMPACT OF REGULARISATION ON SCORES

Figures 12 and 13 shows how model scores on test samples vary as we modify the strength of regularisation. Increasing the strength is seen to favourably impact the scores on the negative class, for both the dominant and rare subgroups. This provides another perspective on why regularisation can be somewhat effective at improving worst-subgroup performance.

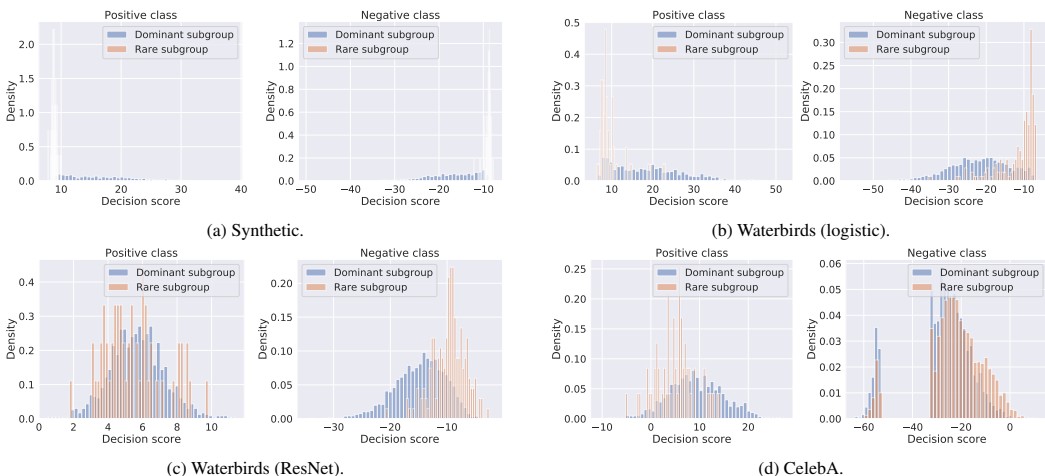

Figure 8: Histograms of model scores on train samples on various datasets, comprising two labels with two sub-groups each. In general, while there are differences in the distributions for the rare and dominant subgroup scores, nearly all such scores are on the correct side of the decision boundary.

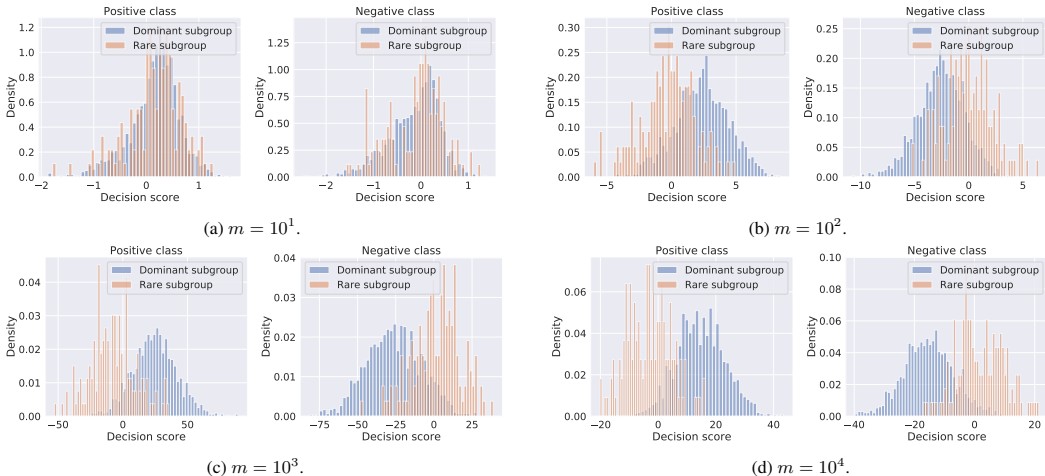

Figure 9: Histograms of model scores on test samples on `synth` as number of projection features $m$ is varied. We see that as the model complexity increases, there is a steady increase in the separation of decision scores between the rare and dominant subgroups for a label. This is in keeping with overparameterisation exacerbating worst-subgroup error.

### D.5 IMPACT OF SUBSAMPLING ON SCORES

Figure 14 shows histograms of model scores on test samples on synthetic dataset, with and without subsampling per Sagawa et al. (2020a). Subsampling is seen to make the scores equitable across the subgroups, which provides another perspective on why this technique can effectively mitigate a bias against rare subgroups.

### D.6 IMPACT OF FRACTION OF RARE SUBGROUPS ON SCORES

The `synth` dataset involves a parameter $p_{\mathrm{dom}}$ in its construction, which controls the relative number of samples belonging to the dominant class. By default, following Sagawa et al. (2020b), we use $p_{\mathrm{dom}} = 0.90$. Figure 15 shows how the tradeoff is affected by changing $p_{\mathrm{dom}}$. As $p_{\mathrm{dom}}$ increases, as expected, it is more challenging to minimise the worst-subgroup error at large $m$. Figure 16 further shows how the test scores for each subgroup are affected by the choice of $p_{\mathrm{dom}}$. As $p_{\mathrm{dom}}$ increases, the rare subgroup scores are seen to significantly diverge from the dominant one.

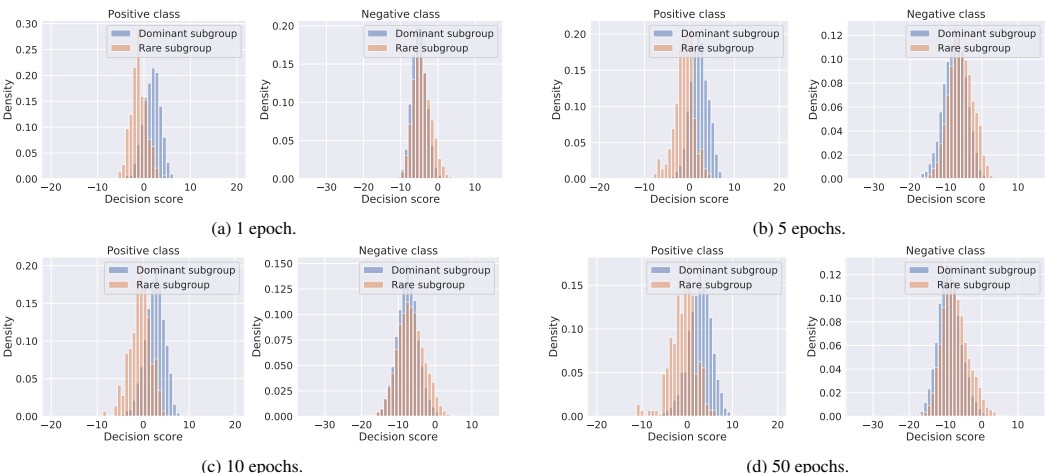

Figure 10: Evolution of histograms of model scores on test samples on `celebA`. The distinction between the scores amongst subgroups of the positive class is visible even after early stopping. With increased training epochs, there is a systematic shift of the negative scores, as the network becomes increasingly confident on them.

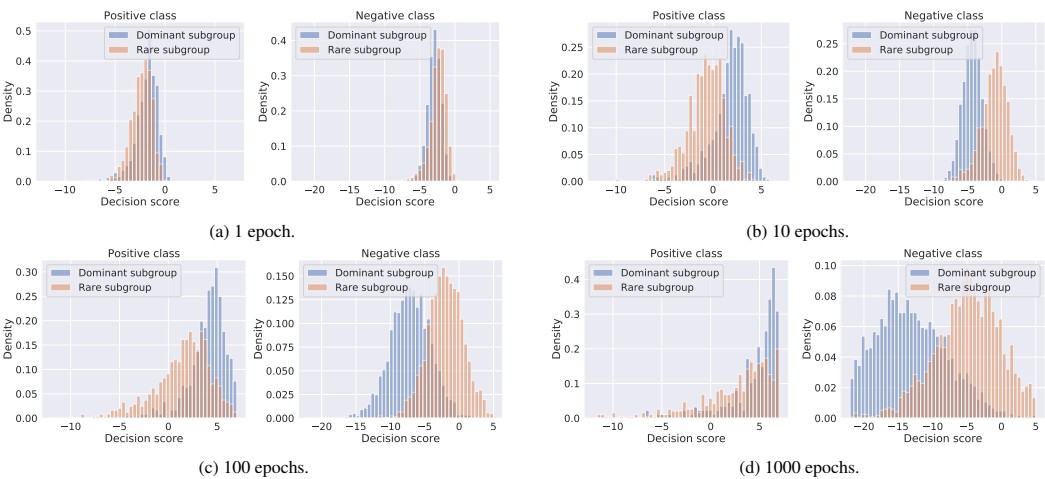

Figure 11: Evolution of histograms of model scores on test samples on `waterbirds`.

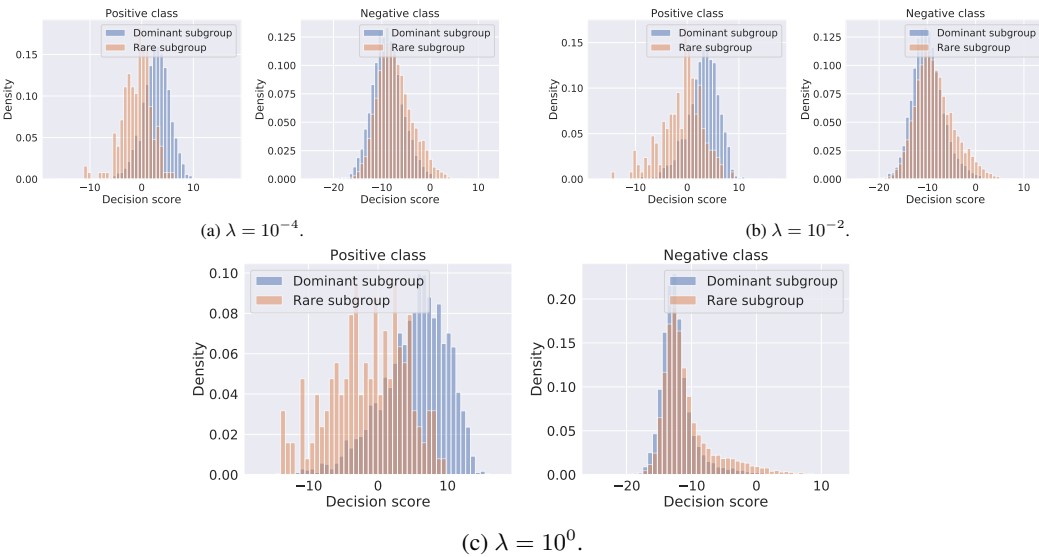

Figure 12: Histograms of model scores on test samples on `celebA` with various strenghts of regularisation. Increasing the strength is seen to favourably impact the scores on the negative class, for both the dominant and rare subgroups.

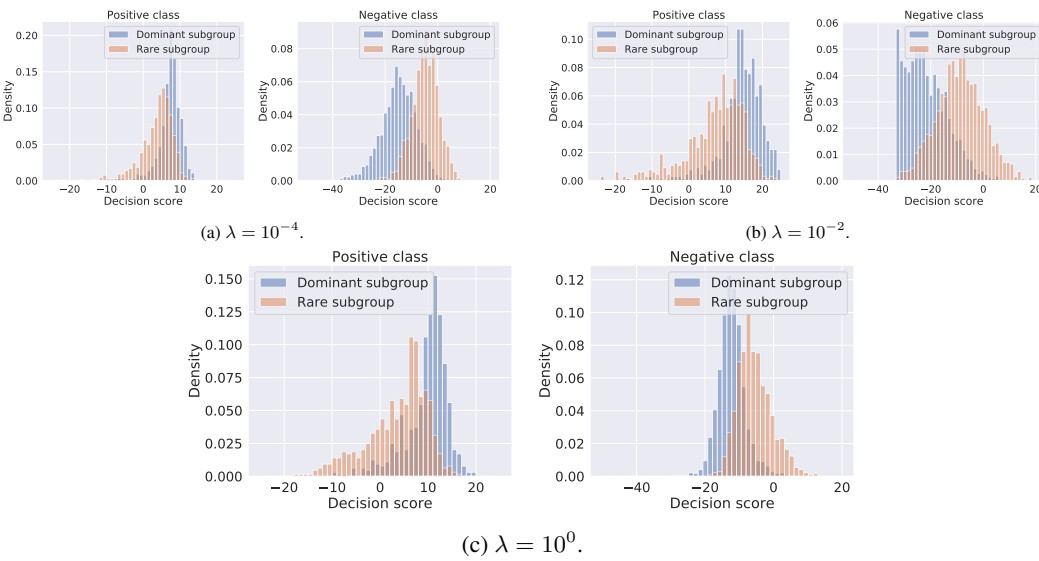

Figure 13: Histograms of model scores on test samples on `waterbirds` with various strengths of regularisation. Increasing the strength is seen to favourably impact the scores on the negative class, for both the dominant and rare subgroups.

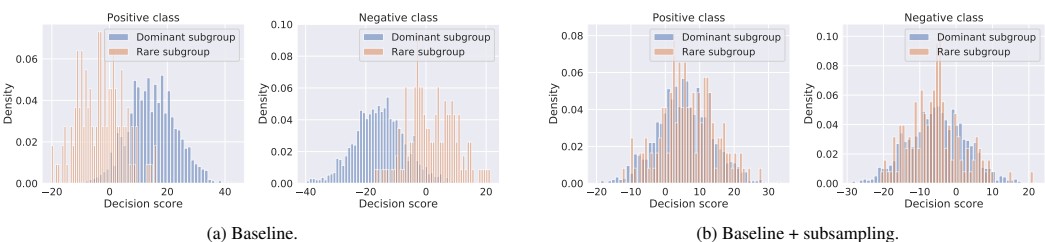

Figure 14: Histograms of model scores on test samples on `synth`, with and without subsampling per Sagawa et al. (2020a). Subsampling is seen to make the scores equitable across the subgroups.

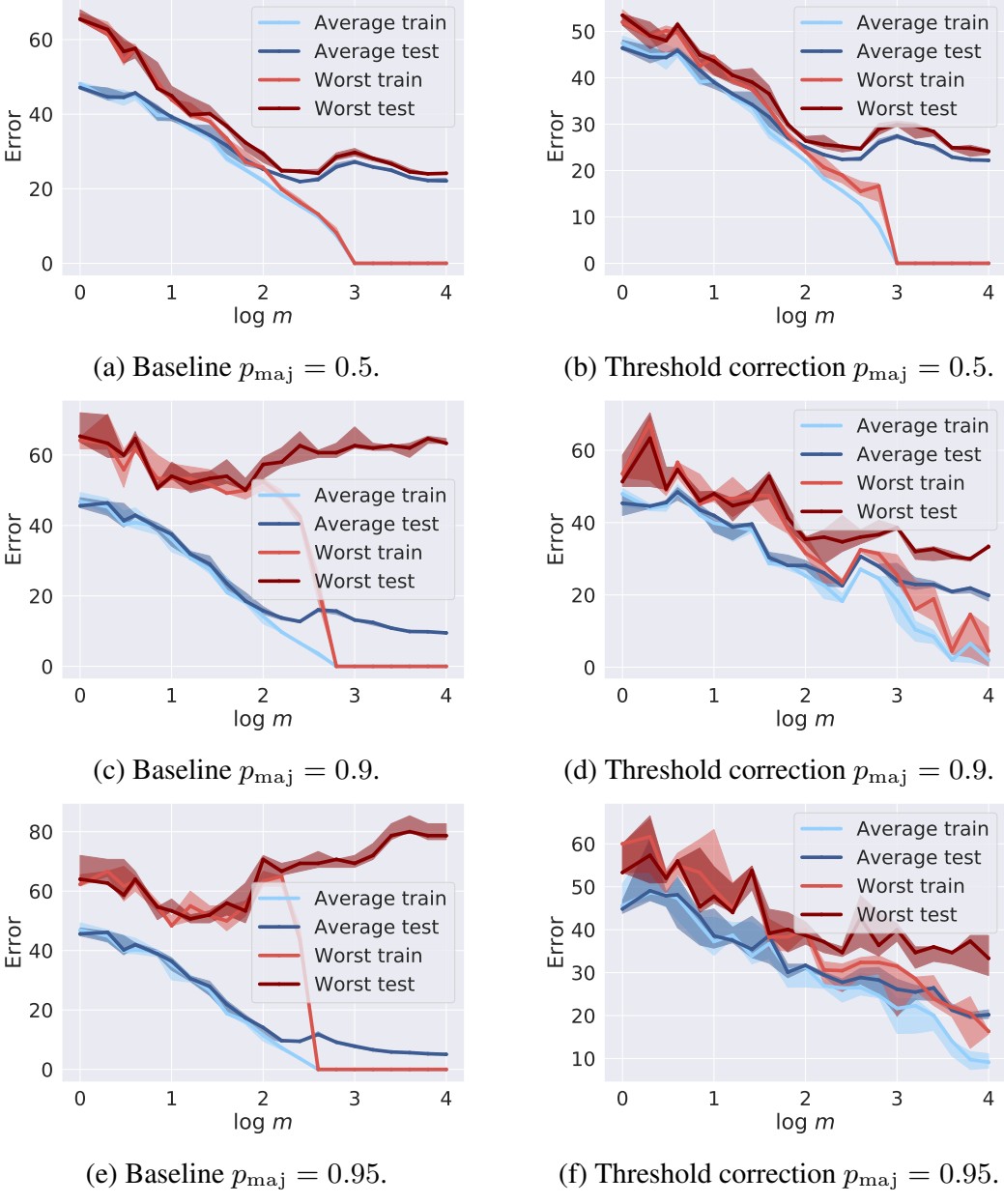

Figure 15: Performance of baseline and threshold correction on `synth`. We vary the fraction of majority samples $p_{\text{maj}}$. This is seen to adversely affect the performance of ERM. However, threshold correction can effectively reduce this error, albeit at the expense of higher variance.

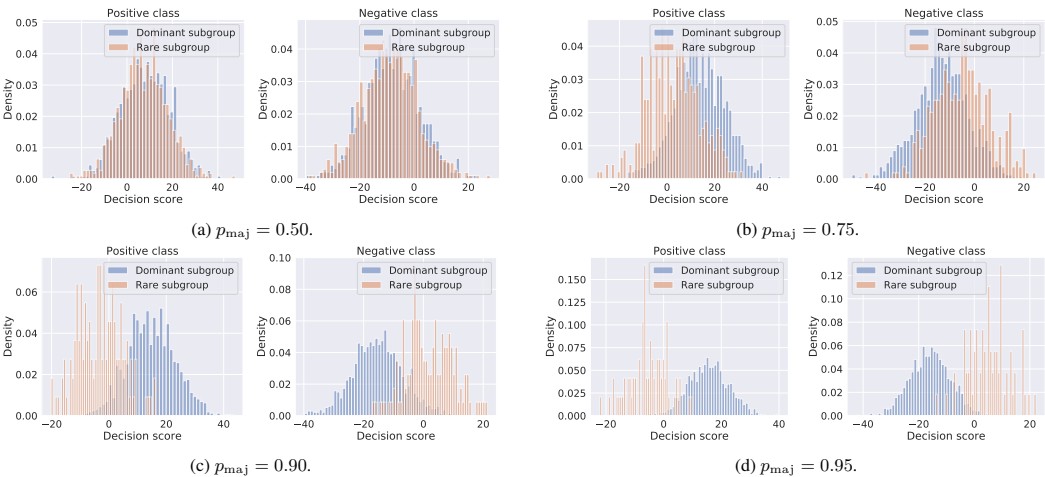

Figure 16: Histograms of model scores on test samples on `synth`, for various values of fraction of majority samples $p_{\mathrm{dom}}$. As this fraction becomes larger, the rare subgroups see progressive shift in their scores compared to the dominant ones.

