# OpenReview forum: "Overparameterisation and worst-case generalisation: friend or foe?"
_ICLR.cc/2021/Conference — ICLR 2021 Poster_

### Official Review · AnonReviewer2 · 2020-10-28
**Nice study of simple methods for improving worst-case subgroup error in overparameterized models**

**Rating:** 7
**Confidence:** 3

**Review:**

This paper studies how to improve the worst-case subgroup error in overparameterized models using two simple post-hoc processing techniques: (1) learning a new linear classification layer of a network, or (2) learning new per-group threshold on the logits. The efficacy of these techniques is evaluated on three synthetic datasets.

Pros:
- The paper studies a timely topic--generalization in overparameterized models-- with some applications to fairness.
- The algorithms presented are very simple and avoid the overhead of more complicated algorithms like DRO or subsampling strategies.
- In the experiments presented, both the thresholding method and the "learn a new classification layer" method significantly improve worst-case group error over ERM and are competitive with DRO.
- The paper is well-written and pleasant to read.

Cons:
- The experimental evaluation is somewhat limited, focusing on three synthetic datasets and two models (ResNet/Logistic Regression). It remains an open question how well the presented technique work more generally both on different model families and on "real" distributions.

Minor:
- There appears to be a discrepancy between the text and the main results table, Table 1. The text says "For example, on celebA, THR reduces the worst-subgroup error from 56.94% to 31.11%", but the table shows THR gets a worst-subgroup error of 12.10. What's going on there?

==============

Update after rebuttal:
Thank you for clarifying the numbers in Table 1 should match the main text. I enjoyed this paper, and I'm keeping my score unchanged.

---

> ### Author Response · Authors · 2020-11-17
> **Response to R2**
>
> Thanks for the detailed feedback and encouraging comments!
>
> *# The experimental evaluation is somewhat limited, focusing on three synthetic datasets and two models (ResNet/Logistic Regression). It remains an open question how well the presented technique work more generally both on different model families and on "real" distributions.*
>
> Please note that these datasets were extensively used in prior work. Per comments to R4, we have added results on more datasets and with more complex group specifications.
>
> *# There appears to be a discrepancy between the text and the main results table, Table 1. The text says "For example, on celebA, THR reduces the worst-subgroup error from 56.94% to 31.11%", but the table shows THR gets a worst-subgroup error of 12.10. What's going on there?*
>
> Thanks for spotting this: this was a typo, which we have fixed.

---

### Official Review · AnonReviewer3 · 2020-10-28
**A demonstration of a few methods for post-hoc improvements for bias in overparametrized models. Interesting, but not sure if there is enough new information.**

**Rating:** 5
**Confidence:** 3

**Review:**

This paper is concerned with potential improvements to the worst-case
(mainly minority class/group) generalizations in over-parametrized
neural networks through post-hoc corrections. The authors demonstrate
the problem and the suggested corrections (that were used in previous
literature) on one artificial classification task as well as two
image classification tasks. The paper shows that post-hoc corrections
may improve the worst-subgroup scores similar to an earlier
state-of-the-art system that modifies the learning objective.

This topic is interesting. The paper is in general written well, and
demonstrates the problem convincingly. The post-hoc fix solution
suggested also seem to be performing reasonably well on the problems /
data sets used in the study.

That being said, it feels the study/paper builds on a few earlier
studies heavily, and I am not fully convinced that there is enough new
findings in the present paper to warrant publication in ICLR.

I also have a few minor notes/suggestions:

- The findings presented in Figure 4 is interesting (mainly the fact
  that the overparametrization seem to be improving the worst-group
  performance with threshold tuning). It would be interesting to see
  more investigation/discussion of what could be the underlying
  reason.

- Page 2 (middle): "also common the fairness literature" ->   "also common in the fairness literature"
- Figure 2 is not very readable. If increasing size/scale is not an
  option due to limited space, taking legend out the figures may also
  improve it. Colorblind friendly colors may also be a good idea.
- There are case (normalization) issues in the references:
  "ml", "t-sne" (not exhaustive, a through check is
  recommended).

---

> ### Author Response · Authors · 2020-11-17
> **Response to R3**
>
> Thanks for the detailed feedback and encouraging comments!
>
> *# That being said, it feels the study/paper builds on a few earlier studies heavily, and I am not fully convinced that there is enough new findings in the present paper to warrant publication in ICLR.*
>
> We reiterate that the novelty of the present work is in showing that :
>
> (1) overparameterised models’ bias on rare subgroups can be in the form of a structured shift in the classification scores, thus deepening the understanding of their behaviour;
>
> (2) this bias can be mitigated via post-hoc modification, thus illustrating that standard training of these models may already capture useful information on rare subgroups.
>
> More broadly, one of our main messages is that the overparameterised setting is not too detached from standard learning with subgroup fairness considerations. As the reviewer notes, this is a timely problem, and so we believe that progress in this regard is of interest. We hope that our study can help guide further explorations in this new field.
>
> *# The findings presented in Figure 4 is interesting (mainly the fact that the overparametrization seem to be improving the worst-group performance with threshold tuning). It would be interesting to see more investigation/discussion of what could be the underlying reason.*
>
> This is one of our central points: the classification layer is biased against rare subgroups. However, increasing model complexity also allows for better modelling (i.e., learned representations) of such samples. This is why, with suitable tuning, one can achieve better performance than in underparameterised settings.
>
> Note also that Sagawa et al. ‘20b already demonstrated that another technique -- subsampling -- can similarly show improvements on worst subgroup performance as model complexity increases. Compared to this, we show that the learned representations from standard training may themselves be sufficient to improve worst subgroup performance.
>
> We have fixed all typos, and updated Figure 2 to have clearer separation from the legend, and use colourblind-friendly colours.

---

### Official Review · AnonReviewer4 · 2020-10-29
**Well-written paper, although a bit limited in overall contributions**

**Rating:** 6
**Confidence:** 3

**Review:**

Summary:
The paper builds upon prior work that shows that overparameterized networks learned by ERM can have poor worst-case performance over pre-defined groups. Specifically, the paper demonstrates that this result is not necessarily due to overparameterized learning poor representations for rare subgroups, but rather mis-calibration in the classification layer that can be addressed with two simple correct techniques: thresholding and re-training the classification layer. They show improvements over ERM in worst-case subgroup error.

Strengths:
1. The paper is very well-written and easy to follow.
2. The paper provides a better understanding of worst-case generalization in overparamaterized models by isolating the issue to the classification layer, which can help machine learning practitioners better understanding how they can address the issue of poor worst-group performance.


Weaknesses:
1. The scope of the work seems largely limited to the set-up from Sagawa a, b. Given that all three datasets are simplified/synthetic (only one attribute, max. 4 subgroups), it would have been great to see how this paper's analysis applied to more complex settings.
2. It is not immediately clear why we would not expect classifier retraining/threshold correction (which as the author notes, are standard techniques) to work for the overparameterized setting? The richness of learned representations is well known, so in some sense the findings are not too surprising, especially given the simple (e.g. binary label) settings that make post-hoc corrections less complex. Could the authors better explain why overparameterization reduces our expectations on the effectiveness of these post-hoc procedures?
3. The requirement of knowing the groups a priori seems rather significant. While one of their main cited works (Sagawa a) seems to have touched on group attribute mis-specification, this was not explored here -- how does having imprecise knowledge of the groups effect performance when using threshold correction or classifier retraining?
4. I have trouble fully understanding Figure 2 without a sense of what "insufficient" or poor representations would look like in a tSNE visualization.


Recommendation:
I recommend acceptance. While I remain concerned about the limited scope of the experiments, I believe the paper adds valuable insights to the overall important topic of robustness / worst-case generalization.

Questions:
1. Does the nature of the attribute (land vs water, or hair color) have any effect on the observed poor worst-group performance, or are the results are mainly due to the fact that some groups are rarer than others? For example, would the authors expect similar results if y=male,female and A=blond,dark for celebA?
2. Did the authors visualize the embeddings of models trained with DRO, to see whether there is any improvement in the learned representations ability to distinguish subgroups?

---

> ### Author Response · Authors · 2020-11-17
> **Response to R4**
>
> Thanks for the detailed feedback and encouraging comments!
>
> *# The scope of the work seems largely limited to the set-up from Sagawa a, b. Given that all three datasets are simplified/synthetic (only one attribute, max. 4 subgroups), it would have been great to see how this paper's analysis applied to more complex settings.*
>
> To show the applicability of our findings in more complex settings, we have added results on:
> 1) CelebA with a larger (64) number of subgroups in Appendix A.1 (see details below).
> 2) a modified version of the MNIST dataset in Appendix A.3, comprising 10 labels and 20 total subgroups. This data is based on the setup of [Goel et al., 2020], and comprises a mixture of standard MNIST and a “corrupted” version with zig-zag images.
>
> In both settings, we find that one can improve on the worst-subgroup error of ERM using the learned representations.
>
> *# It is not immediately clear why we would not expect classifier retraining/threshold correction (which as the author notes, are standard techniques) to work for the overparameterized setting? ... Could the authors better explain why overparameterization reduces our expectations on the effectiveness of these post-hoc procedures?*
>
> As noted in the “Scope and contributions” section (pg 2), the overparameterised setting can defeat techniques that might otherwise work in classical settings, e.g., standard distributionally robust optimisation (c.f. Sagawa et al. ‘20a). Thus, while our proposed idea of decoupling the learned representation and classifier is intuitive in hindsight, its success is not a-priori guaranteed.
>
> In particular, the efficacy of such a technique is unclear given prior work, which established that the predictions of standard training in overparameterised settings leads to systematically biased predictions. These studies do not establish whether or not this bias is solely a function of the classification layer, or in fact pervades the learned representation too.
>
> *# The requirement of knowing the groups a priori seems rather significant. While one of their main cited works (Sagawa a) seems to have touched on group attribute mis-specification, this was not explored here -- how does having imprecise knowledge of the groups effect performance when using threshold correction or classifier retraining?*
>
> We have included in Appendix A.1 results with the setting of imperfectly specified groups, per Sagawa et al. ‘20a, Appendix B. Specifically, we consider A = Wearing Lipstick x Eyeglasses x Smiling x Double Chin x Oval Face, which yields 64 subgroups of Y x A. In this setting, that threshold adjustment based on the subgroup frequencies can similarly improve the worst-subgroup error: this setting yields 16.67% worst-subgroup error, which is a modest increase compared to the 12.10% when using the exact subgroups.
>
> *# I have trouble fully understanding Figure 2 without a sense of what "insufficient" or poor representations would look like in a tSNE visualization.*
>
> In a “poor” representation, samples from the same class may not be clustered together: in particular, rare subgroups from one class may be closer to dominant samples from another class. Of course, this visualisation by itself does not establish the sufficiency of learning with the representations -- it is meant to provide some intuition for why the post-hoc correction techniques of the subsequent section can work.
>
> *# Does the nature of the attribute (land vs water, or hair color) have any effect on the observed poor worst-group performance, or are the results are mainly due to the fact that some groups are rarer than others? For example, would the authors expect similar results if y=male,female and A=blond,dark for celebA?*
>
> We thank the reviewer for this very interesting suggestion. We have included in Appendix A.2 results when swapping the role of Y and A on CelebA. We find that there is an improvement in the baseline performance: the worst-subgroup error goes from 56.94% to 13.67%. Nonetheless, threshold adjustment can further reduce the worst-subgroup error.
>
> In general, we expect that the choice of target Y and spurious attribute A plays an important role in final performance. Indeed, if the target variable is spuriously correlated with many other features in the training set, this could hamper performance. Thus, the problem is not purely one of certain subgroups being rare in the training set.
>
> *# Did the authors visualize the embeddings of models trained with DRO, to see whether there is any improvement in the learned representations ability to distinguish subgroups?*
>
> We have included in Appendix B a plot of the embeddings learned under DRO. We find the separation is visually similar to what is seen under standard ERM. Note however that the slightly improved classification performance under DRO indicates that in the full representation space, there is greater inherent separability of the subgroups.

---

### Comment · ~Thien_Nguyen1 · 2021-11-08
**Implementation Details for Classifier Retraining and Threshold Correction for Reproduceability**

Dear paper's authors. Thank you for the great paper!
I am trying to reimplement/reproduce your Classifier Retraining (CR) and Threshold Correction (TC) method, and I would like to know more details (or wonder if you can make the code public soon?):
1. How do you use the train/val/test split to train the "standard model" in CR and retrain a "distinct classifier" in the retraining phase? i.e. can you elaborate more on "subsample majority group" when retraining the classifier?
2. Details on the retraining phase: what method do you use? Model selection criteria? For how many epochs and with regularization?
3.  Threshold correction: do we require the attribute a(x) in the testing phase to apply the correct threshold?

Any detail (or code) is greatly appreciated. Thank you!

---

> ### Comment · ~Aditya_Krishna_Menon1 · 2021-11-11
> **Re: Implementation Details**
>
> Hi Thien,
>
> Thanks for your interest!
>
> > How do you use the train/val/test split to train the "standard model" in CR and retraining a "distinct classifier" in the retraining phase? i.e. can you elaborate more on "subsample majority group" when retraining the classifier?
>
> The standard model refers to training a network without any modification to the data distribution or loss. The network is trained to minimise the softmax cross-entropy loss on the training split. This involves learning both the embedding and classification layer.
>
> Once this model is trained, we perform random subsampling for the subsequent round of classifier retraining. For the resulting subsampled training set, we take the embeddings learned from the standard model, and fit a linear model (details in next point). Equally, this can be thought of as retraining the network with the embedding layers frozen.
>
> In the subsampling procedure, we partition the training data based on the subgroups. For each subgroup except the smallest ("minority") one, we randomly discard a fraction of the samples so that the resulting group size equals that of the minority subgroup. The resulting dataset will be perfectly balanced amongst the subgroups.
>
> Finally, we perform an additional random subsampling amongst all non-minority samples. This lumps all the samples from the non-minority subgroups together, and picks a random fraction of them. We treat the subsampling fraction as a hyper-parameter, with a fraction of 1.0 meaning we use the balanced dataset created above as-is. We sweep over fractions [ 0.05, 0.1, 0.5, 1.0 ], and pick the one that has the best worst-class performance on the validation set. We then use the model learned with this optimal fraction to make predictions on the test set.
>
> > Details on the retraining phase: what method do you use? Model selection criteria? For how many epochs and with regularization?
>
> We train a linear logistic regression model using scikit-learn. We use the LibLinear solver, which performs full-batch optimisation, with mild (1e-16) L2 regularisation.
>
> > Threshold correction: do we require the attribute a(x) in the testing phase to apply the correct threshold?
>
> Yes, the method assumes access to the attribute a(x) during both training and testing. See also the comments in Section 2, at the top of page 3.

---

> > ### Comment · ~Thien_Nguyen1 · 2022-01-18
> > **Re: Re: Implementation Details**
> >
> > Thank you so much! This clarifies things.

---

### Decision · Program_Chairs · 2021-01-07
**Final Decision**

**Decision:**

Accept (Poster)

**Comment:**

This paper studies how to improve the worst-case subgroup error in overparameterized models using two simple post-hoc processing techniques. All reviewers were positive about the paper, though R5 questioned the novelty of the paper which built heavily on a few previous papers (in particular, it builds heavily on Sagawa et al. 2020a,b). The AC is satisfied with the authors`' response clarifying the novelty. Given that this topic is quite timely and of interest to the ICLR community, and that this paper presented a clean investigation on it, the AC recommends acceptance.